# CANONICAL CORRELATION ANALYSIS WITH IMPLICIT DISTRIBUTIONS

## ABSTRACT

Canonical Correlation Analysis (CCA) is a ubiquitous technique that shows promising performance in multi-view learning problems. Due to the conjugacy of the prior and the likelihood, probabilistic CCA (PCCA) presents the posterior with an analytic solution, which provides probabilistic interpretation for classic linear CCA. As the multi-view data are usually complex in practice, nonlinear mappings are adopted to capture nonlinear dependency among the views. However, the interpretation provided in PCCA cannot be generalized to this nonlinear setting, as the distribution assumptions on the prior and the likelihood makes it restrictive to capture nonlinear dependency. To overcome this bottleneck, in this paper, we provide a novel perspective for CCA based on implicit distributions. Specifically, we present minimum Conditional Mutual Information (CMI) as a new criteria to capture nonlinear dependency for multi-view learning problem. To eliminate the explicit distribution requirement in direct estimation of CMI, we derive an objective whose minimization implicitly leads to the proposed criteria. Based on this objective, we present an implicit probabilistic formulation for CCA, named Implicit CCA (ICCA), which provides a flexible framework to design CCA extensions with implicit distributions. As an instantiation, we present adversarial CCA (ACCA), a nonlinear CCA variant which benefits from consistent encoding achieved by adversarial learning. Quantitative correlation analysis and superior performance on cross-view generation task demonstrate the superiority of the proposed ACCA.

## 1  INTRODUCTION

Canonical correlation analysis (CCA) (Hotelling, 1936) is a powerful technique which has been widely adopted in a range of multi-view learning problems (Haghighi et al., 2008; Kim et al., 2007). Generally, this technique aims at projecting two set of multivariate data into a subspace so that the transformed data are maximally linear correlated. Since classical CCA captures correlations under linear projections, it is represented as linear CCA.

Due to the conjugacy of the prior and the likelihood, probabilistic CCA (PCCA) (Bach & Jordan, 2005) presents analytic solution for the posterior, which provides probabilistic interpretation for linear CCA. Specifically, let $\mathbf{x} \in \mathbb{R}^{d_x}$ and $\mathbf{y} \in \mathbb{R}^{d_y}$ be the random vectors in each view respectively, $\mathbf{z} \in \mathbb{R}^{d_z}$ denotes the latent variable, then the posterior is given as

$$p(\mathbf{z}|\mathbf{x}, \mathbf{y}) = \frac{p(\mathbf{x}, \mathbf{y}|\mathbf{z})p(\mathbf{z})}{p(\mathbf{x}, \mathbf{y})}. \tag{1}$$

To interpret linear CCA, PCCA adopts the following two assumptions: (1) The latent codes $\mathbf{z}$ follows Gaussian distribution; (2) The data in each view are transformed through linear projection. As the prior and the likelihood are conjugate, the posterior $p(\mathbf{z}|\mathbf{x}, \mathbf{y})$ also follows Gaussian distribution, which makes linear correlation an ideal criteria for the multi-view learning problem. In addition, as the two views are both transformed through linear mapping, $p(\mathbf{x}, \mathbf{y}|\mathbf{z})$ can be modeled with the joint covariance matrix, with which the conditional independent assumption is satisfied.

$$p(\mathbf{x}, \mathbf{y}|\mathbf{z}) = p(\mathbf{x}|\mathbf{z})p(\mathbf{y}|\mathbf{z}). \tag{2}$$

Consequently, as maximum likelihood estimation on the joint covariance matrix leads to an analytic solution for the posterior, PCCA addresses the probabilistic interpretation for linear CCA, which greatly deepens the understanding of CCA based models.

However, due to the linear projection, linear CCA can only capture the linear correlation among the views, while the nonlinear dependency that commonly exists in practical multi-view learning

Table 1: Comparison of existing CCA variants. Linear CCA methods are marked with gray, while others are nonlinear CCA extensions. Our proposed ACCA is marked with blue. Column 2 indicates the mapping of the involved variables. Column 3 indicates the adopted dependency criteria. Column 4 indicates whether the model is generative. Column 5 indicates whether the method can handle the distribution with implicit form.

| Methods | Mapping | Dependency criteria | Generative | Implicit | | |
|---|---|---|---|---|---|---|
| | Nonlinear | Nonlinear | | $p(\mathbf{z})$ | $p(\mathbf{x}, \mathbf{y}|\mathbf{z})$ | $p(\mathbf{z}|\mathbf{x}, \mathbf{y})$ |
| CCA | ✗ | ✗ | ✗ | ✗ | ✗ | ✗ |
| PCCA | ✗ | ✗ | ✓ | ✗ | ✗ | ✗ |
| KCCA | ✓ | ✗ | ✗ | ✗ | ✗ | ✗ |
| DCCA | ✓ | ✗ | ✗ | ✗ | ✗ | ✗ |
| CIA | ✗ | ✓ | ✗ | ✗ | ✗ | ✗ |
| LSCDA | ✗ | ✓ | ✗ | ✗ | ✗ | ✗ |
| VCCA | ✓ | - | ✓ | ✗ | ✓ | ✗ |
| Bi-VCCA | ✓ | - | ✓ | ✗ | ✓ | ✗ |
| ACCA | ✓ | ✓ | ✓ | ✓ | ✓ | ✓ |

problems cannot be properly captured. Recently, a stream of nonlinear CCA variants, such as Kernel CCA (KCCA) (Lai & Fyfe, 2000) and Deep CCA (DCCA) (Andrew et al., 2013), were proposed to capture the nonlinear dependency, but the probabilistic interpretation provided in PCCA fails to generalize to these models for mainly two reasons:

1) As nonlinear dependency is to be captured, the Gaussian assumption made on the prior distribution is restricted. Consequently, the linear correlation is no longer an ideal criteria, as the high-order dependency in the latent space cannot be captured.

2) As the mappings to the latent space are complex in these nonlinear variants, conjugacy between the prior and the likelihood is violated. Consequently, the posterior distributions are intractable in these cases, making it restricted to be modelled with Gaussian assumption. Furthermore, the intractability also makes it hard to validate the conditional independent assumption in equation 2 for practical tasks.

To overcome the bottlenecks caused by the assumptions on the prior and the posterior distributions, in this paper, we provide a novel perspective for CCA based on implicit distributions. Specifically, we present minimum Conditional Mutual Information (CMI) as a new criteria for multi-view learning problem, which overcomes the limitations of PCCA. To further eliminate the explicit distribution requirement in the direct estimation of CMI, we derive an objective whose minimization implicitly leads to the proposed criteria. Based on this objective, we present an implicit probabilistic formulation for CCA, named Implicit CCA (ICCA). Proposed ICCA is presented to encompass most of the existing probabilistic CCA variants. It also provides a flexible framework to design new CCA extensions that can handle implicit distributions. As an instantiation, we design adversarial CCA (ACCA) under the ICCA framework. Specifically, ACCA adopts holistic information for the encoding of the data, and adopts adversarial learning scheme to achieve a consistent constraint for the incorporated encodings. This elegant formulation of ACCA enables it to handle CCA problems with implicit distributions. It also leads to the consistent encoding of the involved variables, which benefits ACCA's ability in capturing nonlinear dependency. Extensive experiments verify that ACCA outperforms the baselines in capturing nonlinear dependency between the two views and consequently achieves superior performance in the cross-view generation task.

## 2 RELATED WORK

Recently, various nonlinear CCA variants were proposed to improve the performance of CCA. However, assumptions on the prior and posterior distributions in these models restrict their ability in capturing the nonlinear dependency in real-world multi-view learning problems.

As shown in Table 1, considering the adopted extension strategy, most of the nonlinear CCA variants can be grouped into two categories. For the first category, the nonlinear extension is conducted by capturing linear correlations for nonlinear transformed data. The representative works are Kernel CCA (KCCA) (Lai & Fyfe, 2000) and Deep CCA (DCCA) (Andrew et al., 2013). Although the nonlinear mappings are complex, the common space is still assumed to be Gaussian distributed, as they still adopt the linear correlation as the criteria. As the conjugacy of the prior and the likelihood is violated, this distribution assumption restricted. For the other category, the extension is conducted by capturing high-order dependency for linear transformed data. Most of the existing works adopt mutual information or its variants as the nonlinear dependency measurement, corresponding representative models are Canonical Information Analysis (CIA) (Vestergaard & Nielsen, 2015) and least-squares

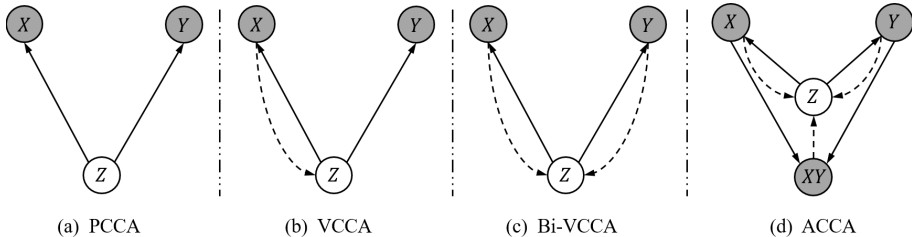

| (a) PCCA | (b) VCCA | (c) Bi-VCCA | (d) ACCA |

Figure 1: Graphical diagrams for generative nonlinear CCA variants. The solid lines in each diagram denote the generative models $p_{\theta_0}(\mathbf{z})p_{\theta_*}(*|\mathbf{z})$. The dashed lines denote the variational approximation $q_{\phi_*}(\mathbf{z}|*)$ to the intractable posterior $p_{\phi_*}(\mathbf{z}|*)$.

canonical dependency analysis (LSCDA) (Karasuyama & Sugiyama, 2012), respectively. However, explicit distributions are still required to estimate the adopted criteria in these methods. The estimation would be extremely complicated for the high dimensional problem. This greatly restricts its ability in handle high-dimensional multi-view learning problem for large datasets.

In (Wang et al., 2016), two generative nonlinear CCA variants, VCCA and Bi-VCCA were proposed, which both can handle implicit likelihood distributions with variational inference. However, the connection between its objective and that of CCA is not obvious. Furthermore, the prior $p(\mathbf{z})$ and the approximate posteriors $q(\mathbf{z}|\mathbf{x},\mathbf{y})$ are still required to be explicit distribution so that the KL-divergence can be computed. However, as the data are usually complex in real-world applications, assumptions on these two distributions restricts the power of these CCA models (Mescheder et al., 2017). For example, if the data in the two views are from two separated clusters, the complex data structure in each view makes it restrictive to make certain distribution assumptions for the latent space.

Drawbacks caused by the restricted distribution assumptions in the existing methods makes is reasonable and promising to propose nonlinear CCA models which can handle implicit distributions.

## 3 IMPLICIT CCA - AN IMPLICIT PROBABILISTIC FORMULATION OF CCA

In this section, we propose an implicit probabilistic formulation for CCA, named Implicit CCA (ICCA). First, we present PCCA and discuss its limitations as preliminary. Next, we present minimum CMI as a new criteria to overcome the limitations of PCCA. Then, we present a general probabilistic formulation of CCA (ICCA) based on the proposed criteria, which provides a flexible framework to design CCA extensions with implicit distributions. Connections between ICCA and existing probabilistic CCA variants are then presented.

### 3.1 PRELIMINARY- FORMULATION OF LINEAR CCA AND PCCA

Let $\mathbf{X} = \{\mathbf{x}^{(i)}\}_{i=1}^N$ and $\mathbf{Y} = \{\mathbf{y}^{(i)}\}_{i=1}^N$ be the inputs with pairwise correspondence in multi-view scenario. Classic linear CCA aims to find linear projections for the two views, $(\mathbf{W}_x'\mathbf{X}, \mathbf{W}_y'\mathbf{Y})$, such that the linear correlation between the projections are mutually maximized.

$$(\mathbf{W}_x^*, \mathbf{W}_y^*) = \arg\max_{\mathbf{W}_x, \mathbf{W}_y} \text{corr}(\mathbf{W}_x'\mathbf{X}, \mathbf{W}_y'\mathbf{Y}) = \arg\max_{\mathbf{W}_x, \mathbf{W}_y} \frac{\mathbf{W}_x'\boldsymbol{\Sigma}_{xy}\mathbf{W}_y}{\sqrt{\mathbf{W}_x'\boldsymbol{\Sigma}_{xx}\mathbf{W}_x\mathbf{W}_y'\boldsymbol{\Sigma}_{yy}\mathbf{W}_y}} \quad (3)$$

Where $\boldsymbol{\Sigma}_{xx}$ and $\boldsymbol{\Sigma}_{yy}$ denote the covariance of $X$ and $Y$ respectively, and $\boldsymbol{\Sigma}_{xy}$ denotes the cross-covariance of $X$ and $Y$. Based on the graphical diagram depicted in Figure 1.(a), PCCA provides probabilistic interpretation for linear CCA by proving that maximum likelihood estimation for the parameters in equation 4 leads to the canonical correlation directions of linear CCA.

$$\mathbf{z} \sim \mathcal{N}(\mathbf{0}, \mathbf{I}_d), \quad \mathbf{x}|\mathbf{z} \sim \mathcal{N}(\mathbf{W}_x\mathbf{z} + \mu_x, \boldsymbol{\Phi}_\mathbf{x}), \quad \mathbf{y}|\mathbf{z} \sim \mathcal{N}(\mathbf{W}_y\mathbf{z} + \mu_y, \boldsymbol{\Phi}_\mathbf{y}), \quad (4)$$

where $\mathbf{W}_x$, $\mathbf{W}_y$, $\boldsymbol{\Phi}_x$, $\boldsymbol{\Phi}_y$, $\mu_x$ and $\mu_y$ are the model parameters; $d$ denotes the dimension of the projected space. As discussed in the introduction, since nonlinear dependency is required to be captured in nonlinear CCA models, PCCA cannot be generalized to these models for two reasons:

- The linear correlation criteria adopted in PCCA cannot capture the high-order dependency between the variables.
- The conditional conditional independent assumption adopted in PCCA cannot be generalized to nonlinear CCA variants.

Consequently, there are two points to consider in probabilistic interpretation of nonlinear CCA variants: (1) a new criteria which can measure nonlinear dependency is required; (2) how to avoid the conditional independent assumption to benefit the generalization ability of the proposed model.

## 3.2 Minimum CMI - a new criteria for multi-view learning problem

In this paper, we present minimum Conditional Mutual Information (CMI) as a new criteria for the CCA-based models, which overcomes the aforementioned limitations of PCCA simultaneously.

Given three random variables $\mathbf{X}$, $\mathbf{Y}$ and $\mathbf{Z}$, CMI, defined in equation 5, measures the expected value of the mutual information between $\mathbf{X}$ and $\mathbf{Y}$ given $\mathbf{Z}$ (Rahimzamani & Kannan, 2017).

$$I(\mathbf{X};\mathbf{Y}|\mathbf{Z}) = \iiint p(\mathbf{z})p(\mathbf{x},\mathbf{y}|\mathbf{z})\log\frac{p(\mathbf{x},\mathbf{y}|\mathbf{z})}{p(\mathbf{x}|\mathbf{z})p(\mathbf{y}|\mathbf{z})}d\mathbf{z}d\mathbf{x}d\mathbf{y} = H(\mathbf{X}|\mathbf{Z}) - H(\mathbf{X}|\mathbf{Y},\mathbf{Z}), \tag{5}$$

where the entropy $H(\mathbf{X}|\mathbf{Z})$ quantifies the uncertainty of $\mathbf{X}$ when $\mathbf{Z}$ is known, and $H(\mathbf{X}|\mathbf{Y},\mathbf{Z})$ quantifies the remaining uncertainty of $\mathbf{X}$ when $\mathbf{Y}$, $\mathbf{Z}$ are both known. where the entropy $H(\mathbf{X}|*)$ quantifies the uncertainty of $\mathbf{X}$ when $*$ is known.

Obviously, CMI overcomes the limitations of PCCA from both the two aspects. First, mutual information (MI) is a general correlation metric which can measure the nonlinear dependency between two random variables (Cover & Thomas, 2012), it is a suitable statistic for interpreting nonlinear CCA variants. Second, measuring CMI of the multi-view inputs given the common latent space avoids the conditional independent assumption of PCCA.

Furthermore, for multi-view learning problem, as the mutual dependency of $\mathbf{X}$ and $\mathbf{Y}$ are maximumly captured in the common latent space $\mathbf{Z}$, further introducing $\mathbf{Y}$ as the condition has not much influence on the uncertainty of $\mathbf{X}$. Thus the difference between $H(\mathbf{X}|\mathbf{Z})$ and $H(\mathbf{X}|\mathbf{Y},\mathbf{Z})$ in equation 5, is to be minimized. Consequently, minimum CMI can be adopted as a new criteria for CCA.

## 3.3 Implicit CCA - an implicit probabilistic formulation of CCA

Based on the proposed minimum CMI criteria, we present an implicit probabilistic formulation for the CCA-based models, named Implicit CCA (ICCA). Specifically, to eliminate the explicit distribution requirement for the estimation of CMI, we derive an objective whose minimization implicitly leads to the proposed criteria. The obtained objective provides a general and flexible framework to instantiate CCA models with implicit distributions.

Here we give the formulation of our model, more details are given in the appendix. Analogous to the derivation of ELBO, the marginal log-likelihood can be rewritten as

$$\log p(\mathbf{X},\mathbf{Y}) = I(\mathbf{X};\mathbf{Y}|\mathbf{Z}) - \mathbb{E}_{\mathbf{x},\mathbf{y}\sim p(\mathbf{x},\mathbf{y})}\mathcal{F}(\mathbf{x},\mathbf{y}), \tag{6}$$

where the first RHS term is the proposed criteria. The second RHS term is the sum over the $\mathcal{F}(\mathbf{x},\mathbf{y})$ for each data pairs. As the model evidence $p(\mathbf{X},\mathbf{Y})$ is a constant with respect to the generative parameters, the minimization of CMI can be achieved by minimizing

$$\min \mathcal{F}(\mathbf{x},\mathbf{y}) = -\mathbb{E}_{\mathbf{z}\sim p(\mathbf{z}|\mathbf{x},\mathbf{y})}[\log p(\mathbf{x}|\mathbf{z}) + \log p(\mathbf{y}|\mathbf{z})] + D_{KL}(p(\mathbf{z}|\mathbf{x},\mathbf{y}) \| p(\mathbf{z})). \tag{7}$$

As the proposed minimum CMI criteria can be implicitly achieved by the minimization of $\mathcal{F}(\mathbf{x},\mathbf{y})$, equation 7 actually provides an implicit probabilistic formulation for CCA-based models, which is named as Implicit CCA (ICCA) in our paper. Obviously, $\mathcal{F}(\mathbf{x},\mathbf{y})$ consists of two components: (1) the reconstruction term, defined by the expectation of data log-likelihood of the two view; (2) the prior regularization term, defined by the KL-divergence of the posterior $p(\mathbf{z}|\mathbf{x},\mathbf{y})$ and the prior $p(\mathbf{z})$.

**ICCA as a framework:** Although proposed ICCA avoids the difficulties in the direct estimation of CMI, the model is still hard to do inference, as the $p(\mathbf{z}|\mathbf{x},\mathbf{y})$ is unknown for practical problems. Thus, efficient approximate inference methods, such as Variational Inference (VI), Bayes Moment Matching (BMM), Generative Adversarial Networks (GANs) or Generative Moment Matching networks (GMMDs) can be adopted to approximate the two terms of ICCA in implementation. Consequently, ICCA provides a flexible framework for designing CCA models with implicit distributions.

## 3.4 Connection between existing probabilistic CCA models and ICCA

The proposed ICCA provides a general framework for the probabilistic formulation of CCA, existing probabilistic CCA variants can be connected to ICCA as follows.

**PCCA:** As illustrated in equation 4, PCCA provides probabilistic interpretation for linear CCA based on two conditions: (1) $\mathbf{x}$ and $\mathbf{y}$ are transformed through a pair of linear projections; (2) both of the likelihoods $p(\mathbf{x}|\mathbf{z})$ and $p(\mathbf{y}|\mathbf{z})$ follow Gaussian distributions. Based on these conditions, PCCA proves that maximum likelihood estimation of the parameters of the model (Figure 1) leads to the solution of the maximum correlation criteria in CCA. Consequently, the PCCA can be interpreted as a special case of ICCA with the aforementioned conditions from the following two aspects:

- Under the intrinsic conditional independent assumption ( equation 2) of PCCA, the minimum CMI criteria of ICCA, is achieved by the solution of PCCA, $I(\mathbf{X}; \mathbf{Y}|\mathbf{Z}) = 0$;
- PCCA adopts maximum likelihood estimation as the objective for CCA, which corresponds to our formulation of ICCA (equation 7).

**VCCA:** As shown in the derivation of VCCA in the appendix B of (Wang et al., 2016), VCCA adopts the approximated posterior from the test view, $q_\phi(\mathbf{z}|\mathbf{x})$ to approximate the true posterior $p(\mathbf{z}|\mathbf{x}, \mathbf{y})$. The objective of VCCA is given as

$$\min \ \mathcal{F}_{q_\phi(\mathbf{z}|\mathbf{x})}(\mathbf{x}, \mathbf{y}) = -\mathbb{E}_{\mathbf{z} \sim q_\phi(\mathbf{z}|\mathbf{x})}[\log p(\mathbf{x}|\mathbf{z}) + \log p(\mathbf{y}|\mathbf{z})] + D_{KL}(q_\phi(\mathbf{z}|\mathbf{x}) \parallel p(\mathbf{z})), \quad (8)$$

For Bi-VCCA, the objective, denoted as equation 9, is defined as a convex combination of the two objectives derived with $q(\mathbf{z}|\mathbf{x})$ and $q(\mathbf{z}|\mathbf{y})$, denoted as

$$\min \ \mathcal{F}_{q_{\phi_1}(\mathbf{z}|\mathbf{x}), q_{\phi_2}(\mathbf{z}|\mathbf{y})}(\mathbf{x}, \mathbf{y}) = \mu \mathcal{F}_{q_{\phi_1}(\mathbf{z}|\mathbf{x})}(\mathbf{x}, \mathbf{y}) + (1 - \mu)\mathcal{F}_{q_{\phi_2}(\mathbf{z}|\mathbf{y})}(\mathbf{x}, \mathbf{y}). \quad (9)$$

Obviously, VCCA and Bi-VCCA are both variational instantiations of ICCA, with different variational inference models to approximate $p(\mathbf{z}|\mathbf{x}, \mathbf{y})$.

## 4 ADVERSARIAL CCA- AN INSTANTIATION OF IMPLICIT ENCODING

To overcome the restrictiveness caused by the assumptions for the incorporated distributions, we specially instantiate Adversarial CCA (ACCA), which can handle CCA problems with implicit distributions, within the proposed ICCA framework. In this section, we first state the motivation for the design of ACCA. Next, we present the techniques adopted for the design of ACCA. Then, we provide a detailed description on the formulation of ACCA.

### 4.1 MOTIVATION

Although VCCA and Bi-VCCA are both variational instantiations of ICCA, which can handle implicit likelihood distributions for multi-view learning problem. Two limitations exist in these models that affect their performance in capturing nonlinear dependency.

1) VCCA and Bi-VCCA assume the common latent space to be Gaussian distributed, which restricted its power in handling complex multi-view learning problems.

*Remark1:* Both VCCA and Bi-VCCA require the prior and approximate posteriors to be explicit Gaussian distributions so that the KL divergence can be computed. However, as the data usually exhibits intractable distributions in real-world applications, assumptions on the data distributions restrict the expressive power of the inference models (Mescheder et al., 2017).

2) VCCA and Bi-VCCA are proposed with inconsistent constraint, which would result in misaligned encodings that affect the inter-view correspondence in the common latent space.

*Remark2:* VCCA adopts $q(\mathbf{z}|\mathbf{x})$ to approximate $p(\mathbf{z}|\mathbf{x}, \mathbf{y})$, with the underling assumption that $q(\mathbf{z}|\mathbf{x}, \mathbf{y}) = q(\mathbf{z}|\mathbf{x})$. Similarly, Bi-VCCA is designed with the assumption that $q(\mathbf{z}|\mathbf{x}, \mathbf{y}) = q(\mathbf{z}|\mathbf{x}) = q(\mathbf{z}|\mathbf{y})$. However, the assumption is hard to be validated, as $q(\mathbf{z}|\mathbf{x}, \mathbf{y})$ is not explicitly modeled. Furthermore, as there is no consistent marginalization for the incorporated inference models, $q(\mathbf{z}|\mathbf{x})$ and $q(\mathbf{z}|\mathbf{y})$, Bi-VCCA potentially suffers from misaligned encoding for the two views, which affects the inter-view correspondence in the common latent space. Consequently, Bi-VCCA would be less effective in cross-view structure output prediction task (Sohn et al., 2015), in which the inter-view correspondence is critical for the performance.

To tackle the aforementioned problems, we specially instantiate Adversarial CCA (ACCA), which achieves consistent encoding for the multi-view inputs, within ICCA framework.

### 4.2 MODEL

As in VCCA and Bi-VCCA, we adopt variational inference to handle implicit likelihood distribution with ICCA. However, two schemes are adopted to overcome the limitations of these two methods.

**Encoding with holistic information**: We first introduce three inference models to provide holistic information for the encoding, which also paves the way for the consistent encoding in ACCA.

Specifically, apart from the encodings, $q(\mathbf{z}|\mathbf{x})$ and $q(\mathbf{z}|\mathbf{y})$, incorporated in Bi-VCCA, we further introduce $q(\mathbf{z}|\mathbf{x}, \mathbf{y})$ that explicitly models the encoding of the joint data distribution $p_d(\mathbf{x}, \mathbf{y})$ for the auxiliary view $\mathbf{XY}$. Graphical model of the proposed ACCA is shown in Figure 1.(d).

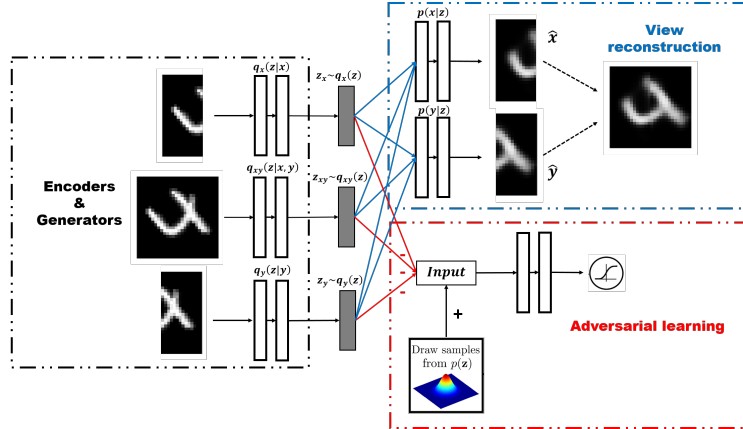

Figure 2: Overall structure of ACCA. The **left panel** corresponds to the encoding with holistic information scheme. For the **right panel**, the top panel represents the view reconstruction, while the bottom panel embodies the adopted adversarial learning scheme, which corresponds to the consistent constraint designed for ACCA.

As the three views provide overall evidence for the inference of $\mathbf{z}$, the three inference models lead to holistic encoding, which benefits the expressiveness of the common latent space. Furthermore, explicitly model $q(\mathbf{z}|\mathbf{x}, \mathbf{y})$ and then approximate all the three encodings avoids the idealistic assumption (*Remank 2*) in Bi-VCCA, thus paving the way for the consistent encoding in ACCA.

**Adversarial learning**: We then adopt an adversarial learning scheme to set a consistent constraint for the three encodings. This also enables ACCA to handle implicit prior and posterior distributions.

Specifically, adopting adversarial learning scheme, each of the three inference models in ACCA defines an aggregated posteriors defined over $\mathbf{z}$ (Makhzani et al., 2015), which can be denoted as

$$q_{\mathbf{x}}(\mathbf{z}) = \int_{\mathbf{x}} q_{\mathbf{x}}(\mathbf{z}|\mathbf{x})\, p_d(\mathbf{x})\, d\mathbf{x}, \qquad q_{\mathbf{y}}(\mathbf{z}) = \int_{\mathbf{y}} q_{\mathbf{y}}(\mathbf{z}|\mathbf{y})\, p_d(\mathbf{y})\, d\mathbf{y},$$
$$q_{\mathbf{x},\mathbf{y}}(\mathbf{z}) = \iint q_{\mathbf{x},\mathbf{y}}(\mathbf{z}|\mathbf{x}, \mathbf{y})\, p_d(\mathbf{x}, \mathbf{y}) d\mathbf{x} d\mathbf{y}, \tag{10}$$

As the posterior $p(\mathbf{z}|\mathbf{x}, \mathbf{y})$ in equation 7 is approximated with these aggregated posteriors, we design generative adversarial network (GAN) that adopts the three inference model as generator and one shared discriminator to drive the approximation of the three aggregated posteriors, denoted as

$$q_{\mathbf{x}}(\mathbf{z}) \approx q_{\mathbf{y}}(\mathbf{z}) \approx q_{\mathbf{x},\mathbf{y}}(\mathbf{z}) \approx p(\mathbf{z}). \tag{11}$$

As the three aggregated posteriors are driven to match the prior $p(\mathbf{z})$ simultaneously via the shared discriminator, the adversarial learning scheme provides a consistent constraint for the three encodings, thus overcomes the misaligned encoding problem in Bi-VCCA.

To conclude, ACCA overcomes the problems in Bi-VCCA from two aspects: 1) As the adversarial learning scheme handles both the prior distribution and the posterior distribution implicitly, proposed ACCA can handle CCA problem with implicit prior distributions. 2) As the holistic encoding provides an overall model evidence for the inference of $\mathbf{z}$, and the adversarial learning scheme drives a consistent approximation of all the marginalized distribution of $\mathbf{z}$, ACCA achieves a consistent encoding for the three inference models, thus tackles the misaligned encoding problem in Bi-VCCA.

### 4.3 FORMULATION OF ACCA

Based on the aforementioned design, the formulation of ACCA within ICCA is given as

$$\begin{aligned} \mathcal{F}_{\mathrm{ACCA}}(\mathbf{x}, \mathbf{y}) \;=\; & -\mathbb{E}_{\mathbf{z}\sim q_{\mathbf{x}}(\mathbf{z})}[\log p(\mathbf{x}|\mathbf{z}) + \log p(\mathbf{y}|\mathbf{z})] - \mathbb{E}_{\mathbf{z}\sim q_{\mathbf{y}}(\mathbf{z})}[\log p(\mathbf{x}|\mathbf{z}) + \log p(\mathbf{y}|\mathbf{z})] \\ & -\mathbb{E}_{\mathbf{z}\sim q_{\mathbf{x},\mathbf{y}}(\mathbf{z})}[\log p(\mathbf{x}|\mathbf{z}) + \log p(\mathbf{y}|\mathbf{z})] + \mathbf{R}_{\mathrm{GAN}}, \end{aligned} \tag{12}$$

where $\mathbf{R}_{\mathrm{GAN}}$ denotes the consistent constraint imposed by the adversarial learning.

Based on the formulation of ACCA, we designed the framework of ACCA. As shown in Figure 2, the framework of ACCA consists of 6 subnetworks, including three encoders, $E_x$, $E_{xy}$ and $E_y$, two decoders $D_x$ and $D_y$, and one adversarial discriminator $\hat{D}$. Description for the role of each subnetworks are summarized in Table 2. Based on the aforementioned architecture, the objective of ACCA is a joint learning problem of the aforementioned components, which is given as

$$\min_{E_x, E_y, E_{xy}, D_x, D_y} \max_{\hat{D}} \; \mathcal{L}_{\mathrm{Recons}}(E_x, E_y, E_{xy}, D_x, D_y) + R_{\mathrm{GAN}}(E_x, E_y, E_{xy}, \hat{D}), \tag{13}$$

Table 2: Description for the roles of each set of subnetworks in ACCA.

| Correspondences in ICCA | Reconstruction | | | Prior regularization |
|---|---|---|---|---|
| **Networks** | $\{E_x, D_x, D_y\}$ | $\{E_y, D_x, D_y\}$ | $\{E_{xy}, D_x, D_y\}$ | $\{E_x, E_y, E_{xy}, \hat{D}\}$ |
| **Description** | Cross-view reconstruction from $E_x(\mathbf{x})$ | Cross-view reconstruction from $E_y(\mathbf{y})$ | Cross-view reconstruction from $E_{xy}(\mathbf{xy})$ | Consistent constraint for the encoders |

where $\mathcal{L}_{\text{Recons}} = \mathcal{L}_{E_x}(E_x, D_x, D_y) + L_{E_y}(E_y, D_x, D_y) + L_{E_{xy}}(E_{xy}, D_x, D_y)$, with each of the three sub-objective corresponds to a variant for the reconstruction term of ICCA.

$$\mathcal{L}_{E_x}(E_x, D_x, D_y) = -\mathbb{E}_{\mathbf{z} \sim q_{\mathbf{x}}(\mathbf{z}|\mathbf{x})} \ [\log(\mathbf{x}|\mathbf{z}) + \log(\mathbf{y}|\mathbf{z})].$$
$$\mathcal{L}_{E_y}(E_y, D_x, D_y) = -\mathbb{E}_{\mathbf{z} \sim q_{\mathbf{y}}(\mathbf{z}|\mathbf{y})} \ [\log(\mathbf{x}|\mathbf{z}) + \log(\mathbf{y}|\mathbf{z})]. \tag{14}$$
$$\mathcal{L}_{E_{xy}}(E_{xy}, D_x, D_y) = -\mathbb{E}_{\mathbf{z} \sim q_{\mathbf{x},\mathbf{y}}(\mathbf{z}|\mathbf{x},\mathbf{y})} \ [\log(\mathbf{x}|\mathbf{z}) + \log(\mathbf{y}|\mathbf{z})].$$

The $R_{\text{GAN}}$ term in equation 13 denotes the consistent constraint imposed by adversarial learning, which corresponds to the prior regularization term in ICCA. The objective is given as

$$\mathcal{R}_{\text{GAN}}(E_x, E_y, E_{xy}, \hat{D}) = \mathbb{E}_{\mathbf{z}' \sim p(\mathbf{z})} \log\left(\hat{D}(\mathbf{z}')\right) + \mathbb{E}_{\mathbf{z}_{\mathbf{x}} \sim q_{\mathbf{x}}(\mathbf{z}|\mathbf{x})} \log\left(1 - \hat{D}(\mathbf{z}_{\mathbf{x}})\right)$$
$$+ \mathbb{E}_{\mathbf{z}_y \sim q_y(\mathbf{z}|\mathbf{y})} \log\left(1 - \hat{D}(\mathbf{z}_{\mathbf{y}})\right) + \mathbb{E}_{\mathbf{z}_{\mathbf{xy}} \sim q_{\mathbf{x},\mathbf{y}}(\mathbf{z}|\mathbf{x},\mathbf{y})} \log\left(1 - \hat{D}(\mathbf{z}_{\mathbf{xy}})\right). \tag{15}$$

## 5 EXPERIMENT

In this section, we first demonstrate the superiority of ACCA in handling implicit distributions with prior specification on artificial toy dataset. Then, we conduct correlation analysis on three real datasets to demonstrate the performance of ACCA in capturing nonlinear dependency. Finally, we specifically demonstrate the effectiveness of consistent encoding in ACCA through alignment verification and cross-view structure output prediction.

**Baselines:** We compare ACCA with several state-of-the-art CCA variants in our experiments. **CCA** (Hotelling, 1936): Linear CCA model that learns linear projections of the two views that are maximally correlated. **Bi-VCCA** (Wang et al., 2016): Bi-deep variational CCA, representative generative nonlinear CCA variants with Gaussian prior on the common latent space. ACCA without the complementary view (**ACCA_NoCV**), variants of ACCA that without the encoding of the complementary view $XY$. Note that, since PCCA requires expensive inference for the posterior, we do not compare with it in the experiment. To demonstrate the effectiveness of handling implicit prior distribution, two versions of ACCA are compared: (1) **ACCA(G)**, ACCA model with standard Gaussian prior. (2) **ACCA(GM)**, ACCA model with Gaussian mixture prior.

**Network setting:** For all the DNN based CCA variants, we model the encoding and decoding functions of the incorporated views with MLP, with a specific number of neurons set in each layer for different datasets. For the discriminator $\hat{D}$, we implement it with a 3layers MLP with the 1024 neurons in each layer. We set the learning rate = 0.001 and epoch = 100 for all the dataset, with the batch size tuned over $\{32, 128, 256, 500, 512, 1000\}$ and the dimension of the latent space $d$ tuned over $\{10, 30, 50, 70\}$ for different datasets. Under the same dimension, the hyperparameters that yield the best HSIC value on the projected tuning set are selected for each method.

**Dependency metric:** Hilbert Schmidt Independence Criterion (HSIC) (Gretton et al., 2005) is a state-of-the-art measurement for the overall dependency among variables. In this paper, we adopt the normalized empirical estimate of HSIC (nHSIC) (Wu et al., 2018) as the metric to evaluate the nonlinear dependency captured in the common latent space. Specifically, we compute the nHSIC for the normalized embedding of the test set obtained with each method. Results of the nHSIC computed with linear kernel and RBF kernel ($\sigma$ set with the F-H distance between the points) are both reported.

### 5.1 PRIOR SPECIFICATION

We first apply ACCA to an artificial dataset generated with specific prior distributions to show that, handling implicit distributions, ACCA could uncover the intrinsics of the latent space which contribute to a better performance in capturing nonlinear dependency.

**Dataset:** Following William (2000), we construct a 3 dimensional toy dataset which posses two proprieties: (1) both the input views follow a complex non-Gaussian distribution; (2) nonlinear dependency exists in these two views. Details for the construction of each dimension of the data are presented in Table 3, where $z$ is from a mixture of three Gaussian distributions, denoted as

$$z \sim \mathcal{N}(0, 1) + 0.8 * \mathcal{N}(2, 0.5) + 1.2 * \mathcal{N}(3, 1.5). \tag{16}$$

Table 3: Details for the design of the toy dataset. The numerals correspond to the coefficients for the linear combination of each dimension in each view. For example, $X_1$, the first dimension of view $\mathbf{X}$ is $(z - 0.3z^2)$.

|  | **X** | | | **Y** | | |
|---|---|---|---|---|---|---|
|  | $X_1$ | $X_2$ | $X_3$ | $Y_1$ | $Y_2$ | $Y_3$ |
| $z$ | 1 | 1 | 0 | 0 | -1 | 1 |
| $z^2$ | -0.3 | 0.3 | 1 | 0 | 0 | 0.3 |
| $z^3$ | 0 | 0 | 0 | 1 | 0.3 | 0 |

The constructed toy dataset totally contains 3000 data of pairs, where 2000 pairs are randomly selected for training, 500 pairs for the validation, and the rest 500 pairs for testing.

We conduct correlation analysis on the embeddings obtained with each method. Note that, for ACCA (GM), the prior is set as equation 16, the true distribution of $\mathbf{z}$. As ACCA handles implicit distributions, which benefits its ability to reveal the intrinsic of the latent space, higher nHSIC is expected to achieve with this method.

Table 4 presents the correlation comparison results on the toy dataset. The table is revealing in several ways: (1) As the embedding space of CCA is as the same dimension as the original space, it achieves the highest nonlinear dependency among the two views. 2) Bi-VCCA obtains the smallest nHSIC value compared with the other generative methods. This indicates that the consistent constraint imposed by adversarial learning benefits the dependency captured in the common latent space. (3) Among the three ACCA variants, ACCA (GM) archives the best result on both settings, which verifies that generative CCA model that can handle implicit distributions can benefit from the flexibility in capturing complex data dependencies.

## 5.2 CORRELATION ANALYSIS

We further test the performance of ACCA in capturing nonlinear dependency on three commonly used real-world multi-view learning datasets.

MNIST handwritten digit dataset (MNIST) (LeCun, 1998) consists of 28 x 28 grayscale digit images where 60,000 of them are for training and 10,000 are for testing. We conduct experiments on two multi-view variants of the MNIST dataset.

**MNIST left right halved dataset(MNIST_LR)** We follow the setting in Andrew et al. (2013) to construct two input views, where view $X$ correspond to the left half of the images and view $Y$ corresponds to the right half of the images. We set the width of the hidden layers of the encoder for $X$, $Y$ and $XY$ to be {2308,1024,1024}, {1608,1024,1024} and {3916,1024,1024}, respectively. 10% images from the training set are randomly sampled for hyperparameter tuning. The dimension of the latent space is set to be 50.

**MNIST noisy dataset (MNIST_Noisy)** We follow the setting in Wang et al. (2016) to construct two input views, where view $X$ corresponds to randomly rotated images and view $Y$ corresponds to noised digits. We set the width of the hidden layers of the encoder $X$,$Y$ and $XY$ to be the same, {1024,1024,1024}. The dimension of the latent space is set to be 30.

**Wisconsin X-ray Microbeam Database (XRMB)** (Westbury, 1994) is a commonly used multi-view speech dataset. We follow the setting in Wang et al. (2014), where view $X$ corresponds to the $273$-$d$ acoustic features and view $Y$ corresponds to the $112$-$d$ articulation features. We use around 1.4M of the utterances for training, 85K for tuning hyperparameters and 0.1M for testing. For the network structure, we follow the setting in Andrew et al. (2013), and set the width of the hidden layers of the encoder $X$,$Y$ and $XY$ to be {1811,1811},{1280,1280} and {3091, 3091}. The dimension of the latent space is set to be 112. Note that, due to the computational issues for the RBF kernel of XRMB, we adopt random feature maps to approximate the original kernel for the computation of nHSIC.

Table 4 presents the correlation comparison for all the methods. We can see that, proposed ACCA, both ACCA(G) and ACCA (GM) achieve superb performance among these methods. Specifically, the dependency captured by ACCA is much better than that of CCA and Bi-VCCA, which indicate that proposed ACCA benefit from consistent encoding and thus better capture the dependency between the two views. In addition, ACCA(G) achieves better results than ACCA_NOCV on most of the settings. This demonstrates that the adopted holistic encoding scheme contributes to the dependency captured in the common latent space.

Table 4: Correlation analysis results on the experimental datasets. The best ones are marked in bold.

| Metric | Datasets | CCA | Bi-VCCA | ACCA_NoCV | ACCA (G) | ACCA (GM) |
|---|---|---|---|---|---|---|
| **nHSIC** (linear kernel) | toy | **0.9999** | 0.9296 | 0.9683 | 0.9581 | 0.9745 |
| | MNIST_LR | 0.4210 | 0.4612 | 0.5233 | 0.5423 | **0.5739** |
| | MNIST_Noisy | 0.0817 | 0.1912 | 0.3343 | 0.3285 | **0.3978** |
| | XRMB | 0.1735 | 0.2049 | 0.2537 | 0.2703 | **0.2748** |
| **nHSIC** (RBF kernel) | toy | **0.9999** | 0.9544 | 0.9702 | 0.9759 | 0.9765 |
| | MNIST_LR | 0.4416 | 0.3804 | 0.5799 | 0.6318 | **0.7387** |
| | MNIST_Noisy | 0.0948 | 0.2076 | 0.2697 | 0.3099 | **0.3261** |
| | XRMB | 0.0022 | 0.0027 | 0.0031 | 0.0044 | **0.0056** |

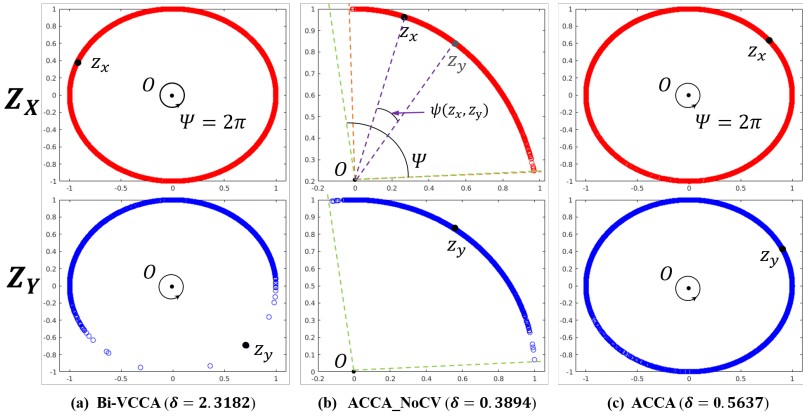

**(a) Bi-VCCA ($\delta = 2.3182$)**   **(b) ACCA_NoCV ($\delta = 0.3894$)**   **(c) ACCA ($\delta = 0.5637$)**

Figure 3: Visualization of the embeddings obtained for the two views, with each row represents the embeddings obtained with view $X$ and view $Y$, respectively. ($\mathbf{z}_x$, $\mathbf{z}_y$) denote a pair of correspondent embedding. $\delta$ indicates the misalignment degree of each method. Methods with smaller value of $\delta$ are better.

## 5.3 VERIFICATION OF CONSISTENT ENCODING

To testify the consistent encoding achieved in the ACCA, we specifically conduct two sets of experiments on the MNIST_LR dataset.

### 5.3.1 ALIGNMENT VERIFICATION

We first embed the data into two-dimensional space to verify the alignment of the multi-view. Specifically, projecting the paired testing data into a Gaussian distributed space, we take the origin $O$ as the reference and adopt the angular difference as the distance measurement for the paired embeddings, which can be denoted as $\phi(\mathbf{z}_x, \mathbf{z}_y) = \angle \mathbf{z}_x O \mathbf{z}_y$ (illustrated with Figure 3.(b)). The degree of the misalignment between the multi-views is measured by

$$\delta = \frac{\sum_1^N \psi(\mathbf{z}_x, \mathbf{z}_y)}{N * \Psi}, \tag{17}$$

Where $\Psi$ denotes the maximum angle of all the embeddings and $N$ is the number of data pairs.

We compare ACCA with Bi-VCCA and ACCA_NoCV as they have the encodings for both the two views. The results of the experiment are presented in Figure 3, in which a larger value of $\delta$ represent a higher degree of misalignment and a inferior performance of the corresponding embedding method.

Obviously, Bi-VCCA suffers from the misaligned encoding problem as regions of this two sets of embeddings are even not overlapped. In addition, the misalignment degree of Bi-VCCA is $\delta = 2.3182$, which is much higher than that of ACCA_NoCV and ACCA. This clearly demonstrates the effectiveness of the consistent constraint imposed by the adversarial learning scheme in ACCA. In addition, the embeddings of ACCA are uniformly distributed in the latent space compared with that of ACCA_NoCV, indicating that the complementary view, $XY$ provide additional information for the holistic encoding, which benefits the effectiveness of the common latent space. Additional experiments for the alignment verification are given in the Appendix.

### 5.3.2 CROSS-VIEW STRUCTURED OUTPUT PREDICTION

To further testify the efficacy of consistent encoding in ACCA, we apply our model to the cross-view structured output prediction task. Specifically, this task aims at whole-image recovery given the partial input image in one of the views. To verify the robustness of ACCA, we divide the test data in each view into four quadrants , and add noise to one, two or three quadrants of the input by

overlapping gray color. Performance of the experiment is evaluated with similarity analysis from both qualitative and quantitative aspects. Since Bi-VCCA is the only baseline method that supports cross-view generation, we compare ACCA with Bi-VCCA in this task.

**Qualitative analysis**: We first evaluate the performance of each method regarding the reality and recognizability of the generated images. Figure 4 presents the visualization of the generated samples for each digit. The figure clearly illustrate that, ACCA generates more realistic images compared to Bi-VCCA, with different levels of noise in the input. For example, given left-half image with 1 quadrant gray color overlaid as input (left subfigure, column 2-4), the generated images by Bi-VCCA are much more blurred and less recognizable than the ones by ACCA (especially in case (a) and (b)). In addition, ACCA can successfully recover the noisy half images which are even confusing for human to recognize. For example, in case (b), the left-half image of digit "5" in the left subfigure looks similar to the digit "4", ACCA succeeds in recovering its true digit. Consequently, overall, ACCA outperforms Bi-VCCA in cross-view structured output prediction task from this perspective.

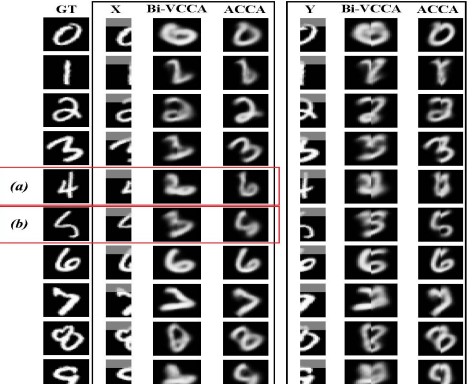 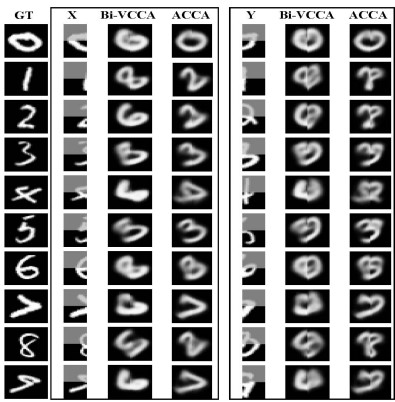

Figure 4: Generated samples given input images with (left) 1 quadrant and (right) 2 quadrants gray color overlaid. In each subfigure, the first column is the ground truth. The next three columns show the input for view $X$ and the corresponding generated image using Bi-VCCA and ACCA, respectively. The last three columns are the input image from view $Y$ and the corresponding generation results of the two methods.

**Quantitative evidence**: We further provide the quantitative evidence by computing the pixel-level accuracy of the generated images to their corresponding original image in Table 5. It clearly demonstrates that our ACCA consistently outperforms Bi-VCCA given input images from different views with different level of noise. What is also interesting in this table is that, using the left-half images as the input tends to generate images more similar to the original images than using the right-half images. It might be because the right-half images contain more information than the left-half images, which will result in a better network training for a more accurate image generation.

Table 5: Pixel-level accuracy for full image recovery given gray color overlaid halved images for different input views on the MNIST_LR dataset.

| Input (halved image) | Methods | Gray color overlaid | | |
|---|---|---|---|---|
| | | 1 quadrant | 2 quadrants | 3 quadrants |
| **Left** | Bi-VCCA | 72.83 | 62.36 | 56.52 |
| | ACCA | **76.85** | **68.95** | **57.63** |
| **Right** | Bi-VCCA | 72.91 | 66.78 | 60.32 |
| | ACCA | **78.31** | **71.30** | **62.77** |

## 6 CONCLUSION

In this paper, we present a theoretical study for CCA based on implicit distributions. Our study discusses the restrictive assumptions of PCCA on nonlinear mappings and provides a general implicit probabilistic formulation for CCA. The proposed framework reveals the connections among different CCA variants and enables flexible design of nonlinear CCA variants with implicit distributions for practical applications. We also propose a consistent encoding strategy in an instantiation of ICCA, which overcomes the misalignment problem in existing generative CCA models. Experiment results verify the superb ability of ACCA in capturing nonlinear dependency, which also contributes to the superior performance of ACCA in cross-view generation task. Furthermore, due to the flexible architecture designed in equation 12, proposed ACCA can be easily extended to multi-view task of $n$ views, with $(n + 1)$ generators and $(n)$ decoders.

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

## 7 APPENDIX

### 7.1 DERIVATION OF ICCA

Analogous to the derivation of ELBO, the expection of the marginal log-likelihood can be rewritten as

$$
\begin{aligned}
\mathbb{E}_{\mathbf{x},\mathbf{y}\sim p(\mathbf{x},\mathbf{y})}\log(\mathbf{x},\mathbf{y}) &= \iint p(\mathbf{x},\mathbf{y})\log p(\mathbf{x},\mathbf{y})d\mathbf{x}d\mathbf{y} \\
&= \iint [\int_{\mathbf{z}} p(\mathbf{x},\mathbf{y},\mathbf{z})\log p(\mathbf{x},\mathbf{y})d\mathbf{z}]d\mathbf{x}d\mathbf{y} \\
&= \iint [\int_{\mathbf{z}} p(\mathbf{z})p(\mathbf{x},\mathbf{y}|\mathbf{z})\log \frac{p(\mathbf{x},\mathbf{y}|\mathbf{z})p(\mathbf{z})}{p(\mathbf{z}|\mathbf{x},\mathbf{y})}d\mathbf{z}]d\mathbf{x}d\mathbf{y} \\
&= \iint [\int_{\mathbf{z}} p(\mathbf{z})p(\mathbf{x},\mathbf{y}|\mathbf{z})\log (\frac{p(\mathbf{x},\mathbf{y}|\mathbf{z})p(\mathbf{z})}{p(\mathbf{z}|\mathbf{x},\mathbf{y})}\cdot\frac{p(\mathbf{z}|\mathbf{x})p(\mathbf{z}|\mathbf{y})}{p(\mathbf{z}|\mathbf{x})p(\mathbf{z}|\mathbf{y})})d\mathbf{z}]d\mathbf{x}d\mathbf{y} \\
&= \iint [\int_{\mathbf{z}} p(\mathbf{z})p(\mathbf{x},\mathbf{y}|\mathbf{z})\log (\frac{p(\mathbf{x},\mathbf{y}|\mathbf{z})p(\mathbf{z})}{p(\mathbf{z}|\mathbf{x},\mathbf{y})}\cdot\frac{p(\mathbf{z}|\mathbf{x})p(\mathbf{z}|\mathbf{y})}{\frac{p(\mathbf{x}|\mathbf{z})p(\mathbf{z})}{p(\mathbf{x})}\cdot\frac{p(\mathbf{y}|\mathbf{z})p(\mathbf{z})}{p(\mathbf{y})}})d\mathbf{z}]d\mathbf{x}d\mathbf{y} \\
&= \iint [\int_{\mathbf{z}} p(\mathbf{z})p(\mathbf{x},\mathbf{y}|\mathbf{z})\log (\frac{p(\mathbf{x},\mathbf{y}|\mathbf{z})}{p(\mathbf{x}|\mathbf{z})p(\mathbf{y}|\mathbf{z})}\cdot\frac{p(\mathbf{z}|\mathbf{x})p(\mathbf{z}|\mathbf{y})p(\mathbf{x})p(\mathbf{y})}{p(\mathbf{z}|\mathbf{x},\mathbf{y})p(\mathbf{z})})d\mathbf{z}]d\mathbf{x}d\mathbf{y} \\
&= \iint [\int_{\mathbf{z}} p(\mathbf{z})p(\mathbf{x},\mathbf{y}|\mathbf{z})\log \frac{p(\mathbf{x},\mathbf{y}|\mathbf{z})}{p(\mathbf{x}|\mathbf{z})p(\mathbf{y}|\mathbf{z})}d\mathbf{z}]d\mathbf{x}d\mathbf{y} \\
&\quad + \iint [\int_{\mathbf{z}} p(\mathbf{z})p(\mathbf{x},\mathbf{y}|\mathbf{z})\log \frac{p(\mathbf{z}|\mathbf{x})p(\mathbf{z}|\mathbf{y})p(\mathbf{x})p(\mathbf{y})}{p(\mathbf{z}|\mathbf{x},\mathbf{y})p(\mathbf{z})}d\mathbf{z}]d\mathbf{x}d\mathbf{y} \\
&= I_{(\mathbf{X},\mathbf{Y})|\mathbf{Z}} + T,
\end{aligned}
\tag{18}
$$

Where

$$
\begin{aligned}
T &= \iint [\int_{\mathbf{z}} p(\mathbf{z})p(\mathbf{x},\mathbf{y}|\mathbf{z})\log \frac{p(\mathbf{z}|\mathbf{x})p(\mathbf{z}|\mathbf{y})p(\mathbf{x})p(\mathbf{y})}{p(\mathbf{z}|\mathbf{x},\mathbf{y})p(\mathbf{z})}d\mathbf{z}]d\mathbf{x}d\mathbf{y} \\
&= \iint [\int_{\mathbf{z}} p(\mathbf{x},\mathbf{y})p(\mathbf{z}|\mathbf{x},\mathbf{y})\log (\frac{p(\mathbf{z}|\mathbf{x})p(\mathbf{z}|\mathbf{y})p(\mathbf{x})p(\mathbf{y})}{p(\mathbf{z}|\mathbf{x},\mathbf{y})p(\mathbf{z})})d\mathbf{z}]d\mathbf{x}d\mathbf{y} \\
&= \iint p(\mathbf{x},\mathbf{y}) [\int_{\mathbf{z}} p(\mathbf{z}|\mathbf{x},\mathbf{y})\log \frac{p(\mathbf{x}|\mathbf{z})p(\mathbf{z})p(\mathbf{y}|\mathbf{z})p(\mathbf{z})}{p(\mathbf{z}|\mathbf{x},\mathbf{y})p(\mathbf{z})}d\mathbf{z}]d\mathbf{x}d\mathbf{y} \\
&= \iint p(\mathbf{x},\mathbf{y}) [\int_{\mathbf{z}} p(\mathbf{z}|\mathbf{x},\mathbf{y})[\log p(\mathbf{x}|\mathbf{z}) + \log p(\mathbf{y}|\mathbf{z})]d\mathbf{z}]d\mathbf{x}d\mathbf{y} \\
&\quad - \iint p(\mathbf{x},\mathbf{y}) [\int_{\mathbf{z}} p(\mathbf{z}|\mathbf{x},\mathbf{y})\log \frac{p(\mathbf{z}|\mathbf{x},\mathbf{y})}{p(\mathbf{z})}d\mathbf{z}]d\mathbf{x}d\mathbf{y} \\
&= \mathbb{E}_{(\mathbf{x},\mathbf{y})\sim p(\mathbf{x},\mathbf{y})}[\mathbb{E}_{\mathbf{z}\sim p(\mathbf{z}|\mathbf{x},\mathbf{y})}[\log p(\mathbf{x}|\mathbf{z}) + \log p(\mathbf{y}|\mathbf{z})] - D_{KL}(p(\mathbf{z}|\mathbf{x},\mathbf{y}) \parallel p(\mathbf{z}))],
\end{aligned}
\tag{19}
$$

Consequently,

$$
\begin{aligned}
\mathbb{E}_{\mathbf{x},\mathbf{y}\sim p(\mathbf{x},\mathbf{y})}\log p(\mathbf{x},\mathbf{y}) &= I(\mathbf{X};\mathbf{Y}|\mathbf{Z}) \\
&\quad + \mathbb{E}_{\mathbf{x},\mathbf{y}\sim p(\mathbf{x},\mathbf{y})}[\mathbb{E}_{\mathbf{z}\sim p(\mathbf{z}|\mathbf{x},\mathbf{y})}[\log p(\mathbf{x}|\mathbf{z}) + \log p(\mathbf{y}|\mathbf{z})]] \\
&\quad - D_{KL}(p(\mathbf{z}|\mathbf{x},\mathbf{y}) \parallel p(\mathbf{z}))
\end{aligned}
\tag{20}
$$

Thus,

$$
I(\mathbf{X};\mathbf{Y}|\mathbf{Z}) = \mathbb{E}_{(\mathbf{x},\mathbf{y})\sim p(\mathbf{x},\mathbf{y})}\log p(\mathbf{x},\mathbf{y}) + \mathbb{E}_{(\mathbf{x},\mathbf{y})\sim p(\mathbf{x},\mathbf{y})}F(\mathbf{x},\mathbf{y}),
\tag{21}
$$

As the $p(\mathbf{X},\mathbf{Y})$ is a constant with respect to the generative parameters, the minimum CMI criteria can be achieved by the minimization of equation 22.

$$
F(\mathbf{x},\mathbf{y}) = -\mathbb{E}_{\mathbf{z}\sim p(\mathbf{z}|\mathbf{x},\mathbf{y})}[\log p(\mathbf{x}|\mathbf{z}) + \log p(\mathbf{y}|\mathbf{z})] + D_{KL}(p(\mathbf{z}|\mathbf{x},\mathbf{y}) \parallel p(\mathbf{z})),
\tag{22}
$$

### 7.2 T-SNE VISUALIZATION FOR THE EMBEDDING OF ACCA AND BI-VCCA

Figure 5, Figure 6 and Figure 7 present the embeddings obtained with each inference model of Bi-VCCA, ACCA_NoCV and ACCA, for MNIST_LR dataset respectively. The results are explainable

from the following aspects: (1) For Bi-VCCA, it is obvious that $\mathbf{z}_y$ fails to show good clustering results. The comparison between that of $\mathbf{z}_x$ indicates that Bi-VCCA suffers from a misalignment encoding for the incorporated two views. (2) For ACCA_NoCV, the cluster results present better alignment for the two views compared with that of Bi-VCCA, which indicates that the adopted adversarial learning scheme benefits the consistent encoding for the two views. 3) For ACCA, we can see that, all the three embeddings show clear clustering structures, and the boundaries are much more clear than that of ACCA_NoCV. This indicates that the adopted holistic encoding scheme also contribute a consistent encoding of the two views, as the incorporated $q(\mathbf{z}|\mathbf{x}, \mathbf{y})$ are forced to reconstruct both the two views.

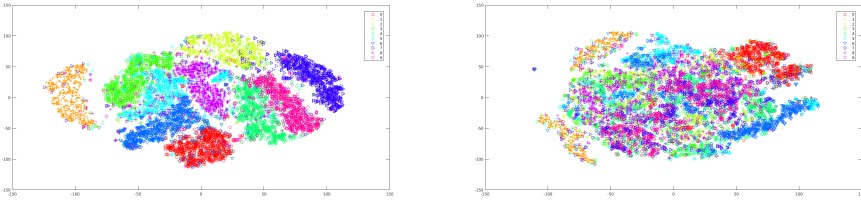

Figure 5: t-SNE visualization of the embeddings of VCCA. **Left**.$\mathbf{z}_x$; **Right**. $\mathbf{z}_y$.

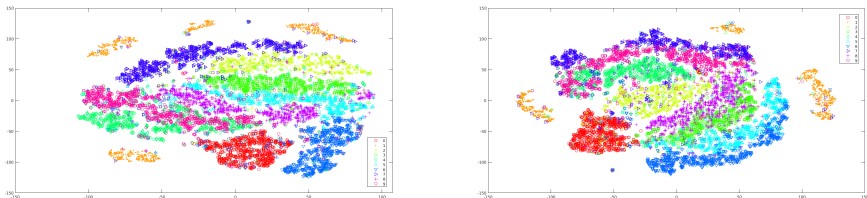

Figure 6: t-SNE visualization of the embeddings of ACCA_NoCV. **Left**.$\mathbf{z}_x$; **Right**. $\mathbf{z}_y$.

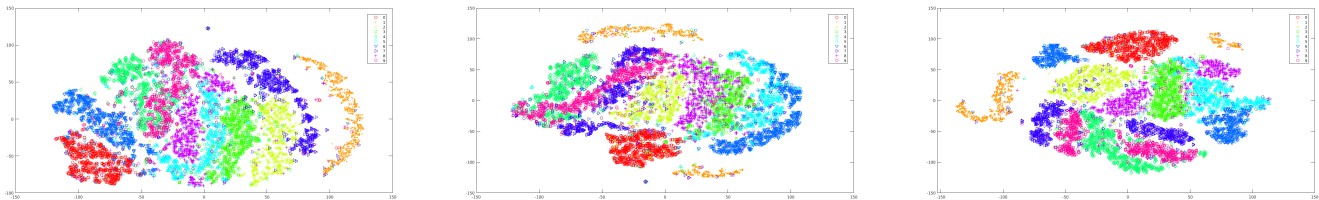

Figure 7: t-SNE visualization of the embeddings of ACCA. **Left**.$\mathbf{z}_x$; **Middle**. $\mathbf{z}_y$; **Right**. $\mathbf{z}_{xy}$.

