# OpenReview forum: "Canonical Correlation Analysis with Implicit Distributions"
_ICLR.cc/2019/Conference_

### Official Review · AnonReviewer1 · 2018-10-14
**The paper does not unambiguously describe the proposed model and algorithm. In the present form, the ICCA framework is not an approach to multi-view learning as it does not construct any transformations of the views. I explain this statement in the review below and, overall, can not recommend this paper for accp.**

**Rating:** 4
**Confidence:** 5

**Review:**

I don't quite see how the proposed approach addresses non-linear canonical correlation analysis. In particular:

1) The main motivation is the minimization of the conditional mutual information I(X;Y|Z), where X and Y correspond to the two views and Z is latent. First of all, what uncertainty does this expression has when X and Y are observations and Z is given? My understanding is that the main objective of any CCA problem should be to find some transformations, say f(X) and g(Y), with some (to be defined) desirable properties. For example, these would correspond to linear transformations, say Ax and By, for classical CCA. Therefore, should not one be interested in minimizing something like I(f(X);g(Y)|Z)?

2) Assuming that the minimization of the conditional mutual information I(X;Y|Z) would be the goal, I don't quite see why the formulation in equation (6) would actually be equivalent (or be some reasonable approximation)?

3) It is well known that differential entropy can be negative (e.g., Cover and Thomas, 2006). Why would the conditional mutual information in equation (4) be non-negative? Alternatively, what would negative values of I(X;Y|Z) mean in the CCA context? My understanding is that one should be interested in minimizing I(X;Y|Z), or its variants with transformations, in *absolute value* to ensure some closeness to conditional independence.

4) Expressions in equation (5)-(6) are general and hold with no assumptions whatsoever for any random variables X, Y, Z (given the expectations/integrals exist). It is therefore not clear what are the variables of this minimization problem? (parameters? but what is the parametric model?)

5) Assuming solving (6) is the goal, this formulation as mentioned by the authors is actually is quite a challenging problem involving latent variables. Some form of this approach explanation would

I can not quite see how the proposed adversarial version would correct or supplement any of these questions.

Other comments:

1) It would be appropriate to cite the probabilistic CCA paper by Bach and Jordan (2005); a better citation for classical CCA would be Hotelling (1936).

2) I find the multiple mentioning of the *non-linear* (in-)dependence confusing. Is this in statistical sense? And how exactly is this related to CCA? Does it have anything to do with the fact that the third and higher order cumulants are zero only for independent variables unless they are Gaussian? Moreover, does this linear independence have any connection with the non-linearity of the proposed CCA approach?

3) What exactly is the *linear correlation criterion* and how does it enter the classical CCA or PCCA formulation (Introduction; bullet point 2)?

4) It would be helpful to introduce the original CCA problem emphasizing that each view, X and Y, are *different* linear transformation of *the same* latent codes z. Moreover, the full description of the models (classical CCA/ PCCA) wouldn't take more than one-two paragraphs and would help the readers to avoid any misunderstanding.

5) Are any assumptions necessary to ensure existence?

---

> ### Author Response · Authors · 2018-11-26
> **Response to Reviewer#1 (Part 1): Thanks for your valuable comments. But there seems to be some misunderstanding on our work and some important details may also be overlooked. We detailedly answered your concerns with the following response.**
>
> Thank you for the constructive comments and suggestions on our work.  We answer your concerns on the proposed ACCA in the following aspects:
> 1).  Our proposed ICCA studies multi-view learning from the generative perspective. (A1)
> 2). The CMI is a theoretically sound criteria to satisfy the requirement for CCA, based on our intention to capture nonlinear dependency with implicit distributions. (A1, A3)
> 3). The feasibility of ICCA and ACCA can be supported by representative works on Deep Generative Models, e.g. VAEs and AAEs. ( A2, A3, A4, A5)
>
> Below, we answer each of your questions detailedly.
>
> A--1). Your concerns on our metric can be answered with two key points of our motivation:  (1). We aim to interpret nonlinear CCA models from the generative perspective; (2). We aim to relax the explicit distribution assumptions on the data for these CCA modeling.
>
> From what we understand here, you interpret the main objective of CCA from the discriminative perspective, which is deviating from our motivations. From the generative perspective, the objective of CCA can be interpreted as learning a compact set of the shared latent variables z that represent a distribution over the observed two-view data x and y (as depicted in Figure1). For this generative model, the latent variable z is to be inferred, instead of “given” as in your understanding.
>
> Actually, $I(f(X), g(Y))$ has been adopted as a metric to capture non-linear dependency (CIA in Table 1). However, explicit distributions are required for f(X) and g(Y), which are intractable to be estimated in these complex expressive nonlinear CCA models. This restricts the model to capture nonlinear dependency (Page 3, Line 1-4) .
>
> Consequently, we present $I(X;Y|Z)$ as the metric, which achieves the following benefits simultaneously: 1). It suits the generative interpretation for multi-view learning problem;  2). It can capture nonlinear dependency implicitly based on the proposed formulation. (Notes: To be detailedly explained with Q2 and Q3).
>
> Note that, although $I(X;Y|Z)$ is the criteria of the proposed ACCA model, the transformations, $f(X)$ and $G(Y)$, are already implicitly implemented in our network (Figure2).
>
> Therefore, the proposed $I(X;Y|Z)$ can be a sound criteria for generative interpretation of nonlinear CCA models.
>
>
> A--2). As we aim to conduct CCA with implicit distributions, following the derivation of ELBO, we derive a surrogate for the proposed $I(X;Y|Z)$ criteria, to eliminate the explicit distribution requirement (Page 3, Line 1-4) for its estimation. As described in Section3.3, we prove that the optimization of Eq. (7) (refer to Eq. (6) in the original version) implicitly leads to $min I(X;Y|Z)$, through the derivation shown in the appendix. The presented derivation can be supported by the ELBO derivation in the variational inference [1].
>
> A--3). Although differential entropy can be negative, the conditional mutual information is always nonnegative based on the Jensen's inequality [2]. Consequently, $ I(X;Y|Z) = 0$ will be the optimal value of the generative CCA problem. Therefore, we do not need to consider the *absolute value* here.
>
> A--4). As explained in Q2, in our paper, Eq.(6) and Eq.(7) (refer to Eq.(5) and Eq.(6) in the original version) are presented to show that connection between the presented $I(X;Y|Z)$ criteria and deduced surrogate objective of the proposed ICCA. (The proof is presented in the appendix)  Practical models are instantiated with different approximate inference methods within the ICCA framework. In these instantiations, the model parameters are what we learn. (Eq.4, Eq.8 and Eq.9).
>
>
> A--5).  As stated in section 4.1, our proposed problem is challenging in two aspects: 1). we study CCA with implicit distributions; 2) we intend to handle task which requires high precision of alignment. Existing methods fail in these two cases.
>
> As illustrated in [3], adversarial training criterion can regularize the aggregated posterior distribution of the latent representation of the autoencoder to arbitrary prior distributions. This kind of approximate inference technique achieves two properties. (1). It allows $q(z|x)$ to act as a deterministic function of x, without explicit assumptions on the posterior distributions.  (2). As the technique drives the approximation of the aggregated posterior to the prior, it achieves a compact latent space in which samples generated from any part of the latent space would be meaningful.
>
> Consequently, in the proposed ACCA, we adopt the adversarial training criterion on the multi-view encodings and adopt a shared discriminator to drive the approximation of these encodings simultaneously. This design enables ACCA to be superior to VCCA in two aspects. (1). ACCA can handle CCA problem with implicit distributions. (2). As ACCA drives the approximation of the three aggregated posteriors to the prior distribution (Eq. 11), it overcomes the misaligned encoding problem in Bi-VCCA (Section 4.2).

---

> ### Author Response · Authors · 2018-11-26
> **Response to Reviewer#1 (Part 2)**
>
> For additional comments:
>
> A--1) Thanks for your suggestion. We have added the reference in the adapted version.
> The paper provides an in-depth probabilistic discussion on linear CCA based on Gaussian assumptions, which greatly deepens the understanding CCA. The restrictions of this work (the bullet points in Page 2) inspired us to re-decide the criteria to deepen the understanding of complex nonlinear CCA models and relax the assumptions.
>
> A--2). In this paper, nonlinear dependency indicates high-order dependency in the statistical sense. For classic CCA, it adopts linear correlation metric to estimate the linear dependency (second-order statistics) between the variables. However, this metric is insufficient to analyze complex practical data which contains higher-order dependency. This is because linear correlation is only an ideal criteria for CCA with Gaussian assumption on the data distributions, which do not hold in general. Consequently, in this work, to capture nonlinear dependency in CCA, we adopt mutual information as the metric, which is a generalized correlation measurement that can handle nonlinear dependency between two random variables.
>
> The non-linearity of the proposed ACCA stems from our motivation to capture nonlinear dependency. Instantiated within ICCA, ACCA also implicitly achieves the $min I(X;Y|Z)$ criteria. For pointwise mutual information for Gaussian distributed data, the proposed criteria is related to linear correlation with the following equation. $ I{(\mathbf{X};\mathbf{Y}|\mathbf{Z})} = \log \frac{1}{1-r^{2}}$, where $r$ denotes the linear correlation between $q(z|x)$ and $q(z|y)$.
>
> However, as we deal with multi-view learning with implicit distributions in ACCA, its connection to the linear independence is inconclusive.
>
>
> A--3). We use “linear correlation criterion ” to indicate correlation measurement which can only estimate the linear dependency between the variables. As it is presented in section 3.1 of the revised version, for CCA and PCCA, the criterion is defined as follows. $\rho = \max\limits_{W_{x},W_{y}} \frac{W_{x}^{'}{\Sigma}_{xy}W_{y}} {\sqrt{W_{x}^{'}{\Sigma}_{xx}W_{x}{W_{y}^{'}{\Sigma}_{yy}W_{y}}}}$, where {\Sigma}_{xx} and {\Sigma}_{yy} denote the covariance of X and Y respectively, and {\Sigma}_{xy} denotes the cross-covariance of X and Y.
>
> A--4). Thanks for your suggestion. We have made revision on the introduction and section 3.1 for clarity.
>
> A--5). Our proposed ACCA do not need any assumptions to ensure existence.
>
> References:
> [1]. D.P. Kingma and Max Welling. Auto-encoding variational bayes. arXiv:1312.6114, 2013.
> [2]. Thomas M Cover and Joy A Thomas. Elements of information theory. John Wiley & Sons, 2012.
> [3]. A. Makhzani, J. Shlensand, N. Jaitly, I. Goodfellow, and B. Frey. Adversarial autoencoders. arXiv:1511.05644, 2015

---

### Official Review · AnonReviewer3 · 2018-11-02
**More explanation of the method can improve the paper**

**Rating:** 6
**Confidence:** 4

**Review:**

This paper proposes to improve deep variational canonical correlation analysis (VCCA, Bi-VCCA) by 1) applying adversarial autoencoders (Makhzani et al. ICLR 2016) to model the encoding from multiple data views (X, Y, XY) to the latent representation (Z); and 2) introducing q(z|x,y) to explicitly encode the joint distribution of two views X,Y. The proposed approach, called adversarial canonical correlation analysis (ACCA), is essentially the application of adversarial autoencoder to multiple data views. Experiments on benchmark datasets, including the MNIST left right halved dataset, MNIST noisy dataset, and Wisconsin X-ray microbeam database, show the proposed ACCA result in higher dependence (measured by the normalized HSIC) between two data views compared to Bi-VCCA.

This paper is well motivated. Since adversarial autoencoder aims to improve based on VAE, it's natural to make use of adversarial autoencoder to improve the original VCCA. The advantage of ACCA is well supported by the experimental result.

In ACCA_NoCV, does the author use a Gaussian prior? If so, could the author provide more intuition to explain why ACCA_NoCV would outperform Bi-VCCA, which 1) also use a Gaussian prior; and 2) also does not use the complementary view XY? Why would adversarial training improve the result?

In ACCA, does the form of the prior distribution have to be specified in advance, such as Gaussian or the Gaussian mixture? Are the parameters of the prior learned during the training?

When comparing the performance of different models, besides normalized HSIC, which is a quite recent approach, does the author compute the log-likelihood on the test set for Bi-VCCA and different variants of ACCA? Which model can achieve the highest test log-likelihood?

According to equation (6), in principle, only q(z|x,y) is needed to approximate the true posterior distribution p(z|x,y). Did the author try to remove the first two terms in the right hand side of Equation (11), i.e., the expectation w.r.t. q_x(z) and q_y(z), and see how the model performance was affected?

Does adversarial training introduce longer training time compared to the Bi-VCCA?

---

> ### Author Response · Authors · 2018-11-26
> **Response to Reviewer#3**
>
> Thanks for your constructive comments and suggestions for improvement of our work. Below, we answer your questions detailedly.
>
> A--1). In ACCA_NoCV, we use Gaussian prior- the same prior as Bi-VCCA. Although it does not use the complementary view, it can outperform Bi-VCCA because the adversarial learning scheme can still provide a consistent marginalization for the two inference models, which alleviates the misaligned encoding problem in Bi-VCCA. Specifically, different from the KL divergence which matches the conditional distribution of z with individual points in each view, adversarial learning drives the approximation of the marginalized distributions of z for the two views (as illustrated with Eq. 10 and Eq. 11). As the variable in each view is marginalized out, this approximation is more robust and can achieve consistent encoding for the two views.
>
> As is presented with section 7.2 in the Appendix, Figure 5 obviously shows that $z_{y}$ of Bi-VCCA fails to show good clustering results. The comparison between that of $z_{x}$ indicates that Bi-VCCA suffers from a misalignment encoding for the incorporated two views. While for ACCA_NoCV in Figure 6,  the clustering result presents better alignment for the two views compared with that of Bi-VCCA, which indicates that the adopted adversarial learning scheme benefits the consistent encoding for the two views.
>
>
> A--2). In the reported experiments, the prior distributions are all set to be Gaussian Mixture and the parameters of the prior are specified in advance. The intention of this setting is to initially verify that better performance would be obtained in generative CCA models when given the suitable prior distribution (In table 4, higher correlation is captured with non‐Gaussian prior). This verifies that ACCA achieves superiority for handling implicit distributions compared with VCCA.
>
> A--3). For the same setting on the nHSIC computation, the average negative log-likelihood achieved with each encoding on the test set of  Bi-VCCA, ACCA_NoCV, ACCA (G), ACCA (GM) are 112.75, 107.26, 94.41 and 103.10 respectively. Consequently, regarding the log-likelihood, ACCA (G) achieves the best result, while Bi-VCCA achieves the worst.
>
> A--4). $q(z|x)$ and $q(z|y)$ are the two principle encodings of CCA. Without them, analysis on the multi-views of CCA, such as correlation analysis and cross-view generation, cannot be conducted.
>
> Actually, the model with the single encoding of $q(z|x,y)$ can be adopted to learn a common embedding for multi-view data, the variant model is worth to be further studied for multi-view embedding task.
>
>
> A--5). The introduction of adversarial training indeed increases the training time of ACCA (1503s vs 2806s), but the increase is tolerable considering its superiority on the result.

---

> > ### Comment · AnonReviewer3 · 2018-12-02
> > **Why constraining the variational posterior q(z|...) be similar to a simple prior p(z)?**
> >
> > I appreciate the authors' detailed response. I like this work and would recommend acceptance. I still have one question on Equation (11).
> >
> > In principle, more flexible posteriors would improve the ELBO of generative models. For example, the focus of black-box variational inference is to design flexible posterior distributions through GAN or normalizing flow.  However, in Equation (11), the variational posterior q(z|x), q(z|y), q(z|x,y) are constrained to be similar to the prior p(z), which is just a simple distribution. Surprisingly, following the current formulation, ACCA with Gaussian prior (rather than Gaussian mixture or more flexible priors) results in the highest test log-likelihood. Would removing the constraint of "q(z|x,y) \approx p(z)"  in Equation (11) improve the log-likelihood in the test set?

---

> > > ### Author Response · Authors · 2018-12-04
> > > **The constraints of ACCA are adopted based on our formulation, but relaxing the constraint indeed improves the log-likelihood in the test set.**
> > >
> > > Thanks for your interest and support.
> > >
> > > Your understanding of our design of ACCA is correct. However, for our ACCA, the constraints are adopted based on our formulation of ICCA ( Eq.7), and it is not “over constrained”.
> > >
> > > Specifically, for Eq.11, the first three terms are required to be satisfied based on the objective of CCA. However, in practical, these three terms are complicated and difficult to be modeled individually.
> > > Inspired by the variational inference, which can drive the approximation of $q(z)~p(z)$, We further adopt specially-designed GAN structure, to drive $q_{x}(z)$, $q_{y}(z)$ and $q_{xy}(z)$ to approximate the same prior $p(z)$ simultaneously. In this way, the approximation of $q_{x}(z)$, $ q_{y}(z)$ and $q_{xy}(z)$ required by CCA, is implicitly satisfied.
> > >
> > > For additional experiments, the log-likelihood obtained without the constraint of "q(z|x,y) \approx p(z)" for ACCA(GM) is $94.912$ (comparable with that of ACCA(G)). This shows that relaxing the constraint improves the log-likelihood in the test set, which coincides with the statement that “flexible posteriors would improve the ELBO of generative models.

---

### Official Review · AnonReviewer2 · 2018-11-04
**Interesting idea but could need more polishing**

**Rating:** 5
**Confidence:** 5

**Review:**

In this paper, the authors attempt to provide a perspective on CCA that is based on implicit distributions.  The authors compare and discuss several variants on CCA that have been proposed over the years, ranging from Linear CCA to Deep CCA and autoencoder variants.  In order to overcome the prior/likelihood distribution assumptions, the authors propose a CCA view that is based on learning implicit distributions, e.g, by using generative adversarial networks.   The authors further motivate their work by comparing with (Bi-)VCCA, claiming that the underlying assumptions lead to inconsistent constraints (or idealistic).  I think the work has merit, and I like the motivation.  Nevetheless, I think stronger experiments are required, as well as improvements in terms of clarity in the writing of the paper, and stronger support for the motivation.   Figure 2 should be better explained in text.  The MNIST experiment is useful, but using GANs usually results in sharper images than say VAE.  Also, comparisons with (i) other models besides Bi-VCCA, and (ii) on other multi-view real-world data (besides the MNIST_LR) would be very useful in terms of communicating the true benefits of this model.

---

> ### Author Response · Authors · 2018-11-26
> **Response to Reviewer#2**
>
> Thanks for your constructive comments and suggestions for improvement of our work.
>
> We have made revisions in the adapted version for a better understanding of our work.
> 1).  Section 1 and section 3 are revised for clarity of our work;
> 2).  Figure1 is further explained in the caption;
>
> Furthermore, our primary objective for MNIST cross-view generation task is to show the multi-view consistency achieved by the proposed ACCA. This VAE generation is consistent with our formulation (section 4.3) and structure design (Figure2) of ACCA. Consequently, the quality of the generated images is not the major consideration in this part. However, we can easily adopt GANs, e. g. VAE- GANs, to improve the quality of the generated images of our model.
>
> We will address your suggestion about the experiments in the future version.

---

### Comment · AnonReviewer1 · 2018-12-02
**Major Flaw**

Hi,

Having read the responses and the paper once more, I don't change my original negative rating. I believe that there is a conceptual flaw which can't be fixed.

Basically, I don't see why equation (6) would be accurate. Indeed, the derivations (equations (17)-(19) in Appendix 7) are provided for the *expectation of the likelihood.* Equation (20) doesn't imply equation (6), where some of the expectations disappear for no reason. Note that equation (6) is the main motivation for the overall approach.

While the authors refer to [1] in their response to my respective question, their derivation in Appendix 7 have nothing to do with the ELBO simply because there is no bound. The procedure they refer to implies a variational approximation to the true posterior in order to lower bound an intractable likelihood. However, the derivations in Appendix 7 have no mention of any variational distribution.

Best,

---

> ### Author Response · Authors · 2018-12-04
> **Thanks for the constructive comments.  We are sorry for the indistinct notations that confused your understanding, but this do not indicates flaw of our motivation.**
>
> For the Left Hand Side (LHS) term of Eq.6,  we improperly adopt $\log p(\mathbf{X}, \mathbf{Y})$ to indicate the expectation of the marginal likelihood (LHS term of Eq.18). The exact form of Eq. 6 would be
>
> $\mathbb{E}_{\mathbf{x},\mathbf{y} \sim p(\mathbf{x},\mathbf{y})} \log {p(\mathbf{x},\mathbf{y})} =
> I{(\mathbf{X};\mathbf{Y}|\mathbf{Z})} - \mathbb{E}_{\mathbf{x},\mathbf{y} \sim p(\mathbf{x},\mathbf{y})} \mathcal{F}(\mathbf{x},\mathbf{y})$
>
> We admit that this is an indistinct notation that confused your understanding.  But this does not influence the soundness of our motivation.
>
> For your second concern, we do not claim that Appendix 7 presents the ELBO. Actually, we simply analogize the derivation of the ELBO in [1], and derive an equality that presents the connection between the expectation of marginal log-likelihood, the CMI and the expected reconstruction error. Based on the derived equality, we obtained a surrogate objective that implicitly leads to the  CMI criteria.
>
> As stated in “ ICCA as a framework ”, since the $p(z|x, y)$ is hard to infer for practical problems, approximate inference methods are to be adopted to instantiate practical models. Formulation of the model would compose of two parts:1). our objective in Eq.7 , with $p(z|x,y)$ substituted by $q(z|*)$; 2). constraint for the approximation  of  $p(z|x, y)$ and $q(z|*)$.
>
> Specifically, if variational inference is adopted for the approximation, the connection between our objective and ELBO can be given as follows.
>
> **********
> \begin{eqnarray}
> 	\mathbb{E}_{\mathbf{x},\mathbf{y} \sim p_{\mathbf{\theta}}(\mathbf{x},\mathbf{y})}\log p_{\mathbf{\theta}}(\mathbf{x},\mathbf{y})
> 	&=& \mathbb{E}_{\mathbf{x},\mathbf{y} \sim p_{\mathbf{\theta}}(\mathbf{x},\mathbf{y})} [\log p(\mathbf{x},\mathbf{y})\int_{\mathbf{z}}q_{\mathbf{\phi}}(\mathbf{z}|\mathbf{x},\mathbf{y}) d\mathbf{z}] \\ \nonumber
> 	&=& \mathbb{E}_{\mathbf{x},\mathbf{y} \sim p_{\mathbf{\theta}}(\mathbf{x},\mathbf{y})} [ \int_{\mathbf{z}}q_{\mathbf{\phi}}(\mathbf{z}|\mathbf{x},\mathbf{y})\log [\frac{q_{\mathbf{\phi}}(\mathbf{z}|\mathbf{x},\mathbf{y})}{{p_{\mathbf{\theta}}(\mathbf{z}|\mathbf{x},\mathbf{y})}} \cdot \frac{{p_{\mathbf{\theta}}(\mathbf{x},\mathbf{y}|\mathbf{z})}}{ p_{\mathbf{\theta}}(\mathbf{x}|\mathbf{z}) p_{\mathbf{\theta}}(\mathbf{y}|\mathbf{z})} \cdot p_{\mathbf{\theta}}(\mathbf{x}|\mathbf{z}) \cdot p_{\mathbf{\theta}}(\mathbf{y}|\mathbf{z}) \cdot \frac{{p_{\mathbf{\theta}}(\mathbf{z})}}{q_{\mathbf{\phi}}(\mathbf{z}|\mathbf{x},\mathbf{y})}] d\mathbf{z}] \\ \nonumber
> 	&=& \mathbb{E}_{\mathbf{x},\mathbf{y} \sim p_{\mathbf{\theta}}(\mathbf{x},\mathbf{y})} D_{KL}(q_{\mathbf{\phi}}(\mathbf{z}|\mathbf{x},\mathbf{y})\parallel p_{\mathbf{\theta}}(\mathbf{z}|\mathbf{x},\mathbf{y})) \\ \nonumber
> 	& & + \iint \ [\int_{\mathbf{z}}p(\mathbf{x},\mathbf{y}){q_{\phi}(\mathbf{z}|\mathbf{x},\mathbf{y})} \log {\frac{p(\mathbf{x},\mathbf{y}|\mathbf{z})}{p(\mathbf{x}|\mathbf{z})p(\mathbf{y}|\mathbf{z})}} d\mathbf{z}] d\mathbf{x}d\mathbf{y}, \\ \nonumber
> 	& &+ \ \mathbb{E}_{\mathbf{x},\mathbf{y} \sim p_{\mathbf{\theta}}(\mathbf{x},\mathbf{y})} [\mathbb{E}_{q_{\mathbf{\phi}}(\mathbf{z}|\mathbf{x},\mathbf{y})} \log [p_{\mathbf{\theta}}(\mathbf{x}|\mathbf{z}) +  \log {p_{\mathbf{\theta}}(\mathbf{y}|\mathbf{z})}] -  D_{KL}{(q_{\mathbf{\phi}}(\mathbf{z}|\mathbf{x},\mathbf{y})\parallel p_{\mathbf{\theta}}(\mathbf{z}))}]
> \end{eqnarray}
>
> **********
>
> We can see that the first RHS term is the KL-divergence that constrains the approximation of
> $p(z|x, y)$ and $q(z|*)$. As $D_{KL}\geq 0$, the rest two terms composes an ELBO of the marginal log-likelihood. The second RHS term can be regarded as the approximation of CMI.  And the third term is our derived objective in Eq.7.
>
> We can see that, if the $D_{KL}=0$ is satisfied, the second term would be CMI, which is guaranteed to be non-negative. This makes our objective an ELBO for the problem. This also indicates that CMI is the criteria that is behind this variational formulation.
>
> The derivation also coincides with our formulation of ACCA, in which we incorporate three $q$ functions to approximate $p(z|x, y)$ through GAN structure. As verified in “correlation analysis” section,  proposed methods captures superior nonlinear dependency compared with other baselines.

---

> > ### Comment · AnonReviewer1 · 2018-12-04
> > **---**
> >
> > In the text above equation (6) you refer to the log-likelihood rather than its expectation. Once you replaced the log-likelihood with its expectation, what kind of statistical estimation/inference technique(s) do you use and why would this imply that *minimization* of CMI is the way to go?

---

> > > ### Author Response · Authors · 2018-12-05
> > > **Thank you very much for pointing out the ambiguous descriptions in our submission.**
> > >
> > > Yes. We admit the typos in the submitted version. The Eq.6 would be consistent with our derivation in Appendix 7.
> > >
> > > The correct form of the equation would be
> > > $\mathbb{E}_{\mathbf{x},\mathbf{y} \sim p(\mathbf{x},\mathbf{y})} \log {p(\mathbf{x},\mathbf{y})} = I{(\mathbf{X};\mathbf{Y}|\mathbf{Z})} - \mathbb{E}_{\mathbf{x},\mathbf{y} \sim p(\mathbf{x},\mathbf{y})} \mathcal{F}(\mathbf{x},\mathbf{y})$
> > >
> > > And the correct explanation in the text would be “the expectation of the marginal log-likelihood”.
> > >
> > > Considering the corrected equation, as supported by the proof in Appendix 7,  the equality presents the connection between the expectation of marginal log-likelihood, the CMI criteria, and the expected reconstruction error. The expectation of $\log p(X, Y)$ would be a constant with respect to the generative parameters”. That is why we claim that “the minimization of CMI can be implicitly achieved by optimizing Eq.7 ”. Consequently, Eq.7 is presented as a surrogate objective that implicitly leads to the CMI criteria.
> > > The connection between Eq.7 and CMI can also be supported by our explanation on the ELBO derivation posted in the previous response.
> > >
> > > Considering  practical tasks, for pairwise data, the expectation would be the average of the marginal log-likelihood on the training set $\mathbb{E}_{\mathbf{x},\mathbf{y} \sim p_{\mathbf{\theta}}(\mathbf{x},\mathbf{y})}\log p_{\mathbf{\theta}}(\mathbf{x},\mathbf{y}) = \frac{1}{N} \log p(\mathbf{X}, \mathbf{Y})$.
> > >
> > > This coincides with our practical objective of ACCA.
> > > $\min  \sum_{i=1}^{N}\frac{1}{N}\mathcal{F}_{\rm ACCA} (\mathbf{x},\mathbf{y})$
> > > where $N$ denotes the number of data pairs. And $\mathcal{F}_{\rm ACCA} (\mathbf{x},\mathbf{y})$ is presented with Eq.12.

---

### Author Response · Authors · 2018-12-08
**Corrections on Eq.(6) and response to the common concerns about our motivation.**

In this post, we first address some misleading typos in our submission, then we give a detailed explanation on the motivation of our work.

First, we express sincere thanks to the AnonReviewer1 for pointing out the misleading typos on Eq.6 in our current submission.

The correct form of the equation would be
$\mathbb{E}_{\mathbf{x},\mathbf{y} \sim p(\mathbf{x},\mathbf{y})} \log {p(\mathbf{x},\mathbf{y})} = I{(\mathbf{X};\mathbf{Y}|\mathbf{Z})} - \mathbb{E}_{\mathbf{x},\mathbf{y} \sim p(\mathbf{x},\mathbf{y})} \mathcal{F}(\mathbf{x},\mathbf{y})$
which is consistent with our consistent with our derivation in Appendix 7.

The correct explanation in the text above this equation would be “the expectation of the marginal log-likelihood”, which can also be regarded as the joint entropy of X, Y.

---

> ### Author Response · Authors · 2018-12-08
> **Part 2: explanation on the motivation of our work.**
>
> Second, the motivation of our work can be explained as follows.
>
> Based on an in-depth discussion on the restrictive assumptions on the probabilistic interpretation of linear CCA (PCCA), we aim to re-decide the criteria to generalize the probabilistic understanding to complex nonlinear CCA models and relax the assumptions.
>
> The CMI criteria:
> We first analyze that minimum CMI is a reasonable criteria that can overcome the limitations of PCCA. Then, we derive an equality (Eq.6) that presents the connection between the expectation of marginal log-likelihood, the CMI and the expected reconstruction error. Based on this equation, we derive a surrogate objective (Eq.7) that can implicitly achieve the proposed minimum CMI criteria. With this objective, the explicit data distribution assumptions are avoided.
>
> The ICCA framework:
> As the $p(z|x, y)$ in Eq.7 is unknown for practical problems, approximate inference methods can be adopted to solve the optimization problem.
> Specifically, If variational inference method is adopted, the ELBO derivation are as follows:
> **********
> \begin{eqnarray}
> \mathbb{E}_{\mathbf{x},\mathbf{y} \sim p_{\mathbf{\theta}}(\mathbf{x},\mathbf{y})}\log p_{\mathbf{\theta}}(\mathbf{x},\mathbf{y})
> &=& \mathbb{E}_{\mathbf{x},\mathbf{y} \sim p_{\mathbf{\theta}}(\mathbf{x},\mathbf{y})} [\log p(\mathbf{x},\mathbf{y})\int_{\mathbf{z}}q_{\mathbf{\phi}}(\mathbf{z}) d\mathbf{z}] \\
> &=& \mathbb{E}_{\mathbf{x},\mathbf{y} \sim p_{\mathbf{\theta}}(\mathbf{x},\mathbf{y})} [ \int_{\mathbf{z}}q_{\mathbf{\phi}}(\mathbf{z})\log [\frac{q_{\mathbf{\phi}}(\mathbf{z})}{{p_{\mathbf{\theta}}(\mathbf{z}|\mathbf{x},\mathbf{y})}} \cdot \frac{{p_{\mathbf{\theta}}(\mathbf{x},\mathbf{y}|\mathbf{z})}}{ p_{\mathbf{\theta}}(\mathbf{x}|\mathbf{z}) p_{\mathbf{\theta}}(\mathbf{y}|\mathbf{z})} \cdot p_{\mathbf{\theta}}(\mathbf{x}|\mathbf{z}) \cdot p_{\mathbf{\theta}}(\mathbf{y}|\mathbf{z}) \cdot \frac{{p_{\mathbf{\theta}}(\mathbf{z})}}{q_{\mathbf{\phi}}(\mathbf{z})}] d\mathbf{z}] \\
> &=& \mathbb{E}_{\mathbf{x},\mathbf{y} \sim p_{\mathbf{\theta}}(\mathbf{x},\mathbf{y})} D_{KL}(q_{\mathbf{\phi}}(\mathbf{z})\parallel p_{\mathbf{\theta}}(\mathbf{z}|\mathbf{x},\mathbf{y})) \\
> & & + \ \textcolor{blue}{\mathbb{E}_{\mathbf{x},\mathbf{y} \sim p_{\mathbf{\theta}}(\mathbf{x},\mathbf{y})} [\mathbb{E}_{q_{\mathbf{\phi}}(\mathbf{z})} \textcolor{red}{\log p_{\mathbf{\theta}}(\mathbf{x},\mathbf{y}|\mathbf{z})} -  D_{KL}{(q_{\mathbf{\phi}}(\mathbf{z})\parallel p_{\mathbf{\theta}}(\mathbf{z}))}] \Rightarrow  \mathcal{L}_{1}(\mathbf{x},\mathbf{y};\mathbf{\theta},\mathbf{\phi})} \\
> &=& \mathbb{E}_{\mathbf{x},\mathbf{y} \sim p_{\mathbf{\theta}}(\mathbf{x},\mathbf{y})} D_{KL}(q_{\mathbf{\phi}}(\mathbf{z})\parallel p_{\mathbf{\theta}}(\mathbf{z}|\mathbf{x},\mathbf{y})) \\
> & & + \iint \ [\int_{\mathbf{z}}p(\mathbf{x},\mathbf{y}){q_{\phi}(\mathbf{z}|\mathbf{x},\mathbf{y})} \log {\frac{p(\mathbf{x},\mathbf{y}|\mathbf{z})}{p(\mathbf{x}|\mathbf{z})p(\mathbf{y}|\mathbf{z})}} d\mathbf{z}] d\mathbf{x}d\mathbf{y}, \\
> & &+ \ \mathbb{E}_{\mathbf{x},\mathbf{y} \sim p_{\mathbf{\theta}}(\mathbf{x},\mathbf{y})} [\mathbb{E}_{q_{\mathbf{\phi}}(\mathbf{z})} \log [p_{\mathbf{\theta}}(\mathbf{x}|\mathbf{z}) +  \log {p_{\mathbf{\theta}}(\mathbf{y}|\mathbf{z})}] -  D_{KL}{(q_{\mathbf{\phi}}(\mathbf{z})\parallel p_{\mathbf{\theta}}(\mathbf{z}))}]
> \end{eqnarray}
> **********
> For the last equation, we can see that based on the KL-divergence that constrains the approximation of $p(z|x, y)$ and $q(z)$, The second RHS term can be regarded as the approximation of CMI. Consequently, adopting variational inference for the approximation, our objective (substituting $p(z|x,y)$ with $q(z)$) can implicitly achieve the minimum CMI criteria proposed for CCA.
>
> Furthermore, considering the difference between the last equation and $ \mathcal{L}_{1}(\mathbf{x},\mathbf{y};\mathbf{\theta},\mathbf{\phi})} $, it is verified that with the adopted CMI criteria, the conditional independent assumption is avoided.
>
> The ACCA instantiation:
> Considering the limitation of misaligned encoding of VCCA, we design ACCA with adversarial learning technique based on Eq.11.
> Specifically,  apart from the first two terms that are naturally required by the objective of CCA, We further introduce $q(z|x, y)$ to provide holistic information for the joint data distribution $p(x, y)$. Then, considering the difficulty in modeling these approximations individually, we constrain all these three encodings, $q(z|x)$ $q(z|y)$ and $q(z|x, y)$, to be similar to the prior p(z). In this way, the approximation of these encodings is implicitly satisfied.
>
> Based on the aforementioned analysis, proposed ACCA is formulated with Eq. 12, which conforms with both our formulation of ICCA in Eq.7 and the network design in Figure 2.
>
> In the end, thanks again for all the reviewers’ valuable comments and constructive suggestions on our work. we will strive to make this work a better one.

---

### Meta-Review · Area_Chair1 · 2018-12-14

**Confidence:** 4
**Recommendation:** Reject

**Metareview:**

This manuscript proposes an implicit generative modeling approach for the non-linear CCA problem. One contribution is the proposal of Conditional Mutual Information (CMI) as a criterion to capture nonlinear dependency, resulting in an objective that can be solved using implicit distributions. The work seems to be well motivated and of interest to the community.

The reviewers and AC opinions were mixed, and the rebuttal did not completely address the concerns. In particular, a reviewer pointed out an issue with a derivation in the paper, and the issue was not satisfactorily resolved by the authors. Some additional reading suggests that the misunderstanding may be partially due to incomplete notation and other issues with clarity of writing.